**PLOS | COMPUTATIONAL BIOLOGY**

# BrainIAK tutorials: User-friendly learning materials for advanced fMRI analysis

**Manoj Kumar**[1]*, **Cameron T. Ellis**[2], **Qihong Lu**[1], **Hejia Zhang**[3], **Mihai Capotă**[4], **Theodore L. Willke**[4], **Peter J. Ramadge**[1,3], **Nicholas B. Turk-Browne**[2], **Kenneth A. Norman**[1,5]

**1** Princeton Neuroscience Institute, Princeton University, Princeton, New Jersey, United States of America, **2** Department of Psychology, Yale University, New Haven, Connecticut, United States of America, **3** Center for Statistics and Machine Learning, Princeton University, Princeton, New Jersey, United States of America, **4** Brain-Inspired Computing Lab, Intel Corporation, Hillsboro, Oregon, United States of America, **5** Department of Psychology, Princeton University, Princeton, New Jersey, United States of America

\* mk35@princeton.edu

## Abstract

Advanced brain imaging analysis methods, including multivariate pattern analysis (MVPA), functional connectivity, and functional alignment, have become powerful tools in cognitive neuroscience over the past decade. These tools are implemented in custom code and separate packages, often requiring different software and language proficiencies. Although usable by expert researchers, novice users face a steep learning curve. These difficulties stem from the use of new programming languages (e.g., Python), learning how to apply machine-learning methods to high-dimensional fMRI data, and minimal documentation and training materials. Furthermore, most standard fMRI analysis packages (e.g., AFNI, FSL, SPM) focus on preprocessing and univariate analyses, leaving a gap in how to integrate with advanced tools. To address these needs, we developed BrainIAK (brainiak.org), an open-source Python software package that seamlessly integrates several cutting-edge, computationally efficient techniques with other Python packages (e.g., Nilearn, Scikit-learn) for file handling, visualization, and machine learning. To disseminate these powerful tools, we developed user-friendly tutorials (in Jupyter format; https://brainiak.org/tutorials/) for learning BrainIAK and advanced fMRI analysis in Python more generally. These materials cover techniques including: MVPA (pattern classification and representational similarity analysis); parallelized searchlight analysis; background connectivity; full correlation matrix analysis; inter-subject correlation; inter-subject functional connectivity; shared response modeling; event segmentation using hidden Markov models; and real-time fMRI. For long-running jobs or large memory needs we provide detailed guidance on high-performance computing clusters. These notebooks were successfully tested at multiple sites, including as problem sets for courses at Yale and Princeton universities and at various workshops and hackathons. These materials are freely shared, with the hope that they become part of a pool of open-source software and educational materials for large-scale, reproducible fMRI analysis and accelerated discovery.

**Data Availability Statement:** The data are publicly available and hosted on Zotero: https://doi.org/10.5281/zenodo.2598755.

**Funding:** Funding for this project was provided by Intel Labs (https://www.intel.com/intellabs) to P.J.

R., N.B.T.-B, and K.A.N. The funders had no role in study design, data collection and analysis, or decision to publish. M.C. and T.L.W. were involved in the development of the BrainIAK package and in preparation of the manuscript as noted in the author contributions.

**Competing interests:** The authors have declared that no competing interests exist.

## Author summary

The analysis of brain activity, as measured using functional magnetic resonance imaging (fMRI), has led to significant discoveries about how the brain processes information and how this is affected by disease. However, exhaustive multivariate analyses in space and time, run across a large number of subjects, can be complex and computationally intensive, creating a high barrier for entry into this field. Furthermore, the materials available to learn these methods do not encompass all the methods used, work is often published with no publicly available code, and the analyses are often difficult to run on large datasets without cluster computing. We have created interactive software tutorials that make it easy to understand and execute advanced analyses on fMRI data using the BrainIAK package—an open-source package built in Python. We have released these tutorials freely to the public and have significantly reduced computational roadblocks for users by making it possible to run the tutorials with a web browser and internet connection. We hope that this facilitated access and the usability of the underlying code—a compendium for how to program and optimize the latest fMRI analyses—will accelerate training, reproducibility, and discovery in cognitive neuroscience.

This is a *PLOS Computational Biology* Software paper.

## Introduction

The latest methods for analyzing brain activity recorded via functional magnetic resonance imaging (fMRI) are complex to learn and execute. This is particularly true for multivariate pattern analysis (MVPA) methods, which focus on extracting information about a person's cognitive state (i.e., percepts, thoughts, memories) from spatially and/or temporally distributed patterns of fMRI activity. Beginners and even intermediate users face a steep learning curve and uncertainty in using these complex techniques. Even expert users are hesitant to add new, more advanced techniques to their existing pipelines, and face significant software and hardware challenges in doing so. These difficulties continue, even though MVPA has been used successfully for almost two decades to answer fundamental questions in cognitive neuroscience.

MVPA encompasses a wide range of analyses: from pattern classifiers that map between distributed brain patterns and cognitive states [1–4], to techniques that explore the similarity structure exploited by classifiers (e.g., representational similarity analysis, RSA; [5,6]). There are also related multivariate techniques for functional connectivity and functional alignment, including: full correlation matrix analysis (FCMA; [7]), inter-subject correlation (ISC; [8,9]), inter-subject functional connectivity (ISFC; [10]), shared response modeling (SRM; [11]), and event segmentation [12]. These analyses can be run after data collection is complete or in real-time for neurofeedback training or adaptive design optimization [13–16].

There exist multiple open-source packages that implement MVPA and RSA techniques. Some of these packages require commercial software such as MATLAB with paid licenses and proprietary code (e.g., Princeton MVPA Toolbox, The Decoding Toolbox [17], and CoSMoMVPA [18]), while others are completely open-source (e.g., Nilearn [19] and PyMVPA [20,21]). Although all of these packages cover a broad range of MVPA and RSA techniques,

they do not cover techniques such as FCMA, ISC, ISFC, SRM, and event segmentation. One barrier to increasing the accessibility of these techniques is that, in most cases, they were created as custom code within individual labs and are thus not part of other fMRI software analysis packages. To address this, we implemented and released them in an open-source Python package called BrainIAK. The tutorials that are the focus of this article provide extensive background, code, and exercises, which serve as structured guidance for learning how to use these and other advanced fMRI analysis techniques. The tutorials also show how to use methods in other packages such as Nilearn and integrate them with the methods in BrainIAK.

In a typical fMRI analysis pipeline, the data are first pre-processed, a general linear model (GLM) might be fit, and then MVPA or other more advanced analyses are performed. For pre-processing and GLM analysis of fMRI data, a number of tutorials and bootcamps are available to learn software packages such as AFNI, FSL, SPM, and fmriprep [22–25]; a recent publicly released course at http://dartbrains.org also nicely covers these topics. In contrast, for MVPA and more advanced analyses, fewer educational materials are available. We designed the present tutorials to make it easier for the novice user to learn these techniques. An expert user can use our materials to understand BrainIAK's implementation of these techniques, to train other researchers, and to teach research methods classes.

There are three main steps to learning and implementing BrainIAK methods: (1) learning to write code and scripts, (2) understanding machine learning algorithms and how to apply them to cognitive neuroscience data, and (3) executing jobs on high-performance compute clusters. We elaborate on each of these steps below.

First, one needs to learn a programming language; for example, BrainIAK uses Python. This can present a significant challenge to a beginner as learning to program and how to apply these skills to scientific computing is a time-consuming process. Such skills have only recently been added to the curriculum in some psychology and neuroscience departments, and been included as components of hackathons and summer schools. As instructors tend to teach in the language they are most familiar with, different programming languages are often used to teach various techniques, making it difficult for users to switch flexibly between methods.

Second, the analysis techniques in BrainIAK involve extensive use of machine learning algorithms that may be unfamiliar to cognitive neuroscientists. There are multiple tutorials on machine learning available (see examples on Scikit-learn https://scikit-learn.org/stable/auto_examples/index.html); however, only a few cover the use of machine learning in cognitive neuroscience: for example, the documentation for Nilearn [19], lectures from the MIND summer school, lectures from the Organization for Human Brain Mapping education section and hackathons, and blogs such as MVPA Meanderings. For many of the cutting-edge techniques in BrainIAK, no tutorials exist (one notable exception is the volumetric searchlight technique; tutorials for this method are included in the PyMVPA [20,21] and Nilearn [19] packages) or they are taught only as a part of special workshops. Furthermore, the application of general-purpose machine learning algorithms in cognitive neuroscience needs to be done with care, as not all data are independent of each other in space or time; this has led to the insidious problem of circular inference or "double dipping" [26].

Third, the execution of these programs on high-performance compute clusters is non-trivial even for advanced practitioners who are proficient at executing code on individual machines. Using clusters can accelerate analyses dramatically through parallelization, but sizing the memory needed and enabling parallel code execution for optimal run-times requires an understanding of how jobs are scheduled and processed in a cluster environment. It is a challenge to find training materials on how to run fMRI analyses on a compute cluster, although, resources are becoming increasingly available, for example, lectures on

Neurohackademy (https://neurohackademy.org/course_type/lectures/); and forums such as NeuroStars (https://neurostars.org) for using fmriprep on clusters [25].

We have created learning materials (herein referred to as tutorials) that address each of the above challenges, making it easier for novice users to learn MVPA and for expert users to learn more advanced BrainIAK analysis techniques, such as FCMA and SRM. To aid learning to code, the tutorials provide an interactive environment to read, write, and execute Python. Specifically, for novice users, a simple way to learn programming is to study small snippets of code with a clear description of what is being accomplished by the code. Our use of Jupyter notebooks [27] allows for detailed explanations of the code with text and figures embedded in-line. The user can execute the code step-by-step and interact with data at each step using plotting functions. In order to ease users into the use of advanced analysis techniques, we first introduce them to a fully-working but simplified version of the code. After mastering this version, we encourage users to delve deeper and learn more about helper functions and input/output variables. Expert users, who may wish to examine the details of how the data are being processed, or modify the code to suit their needs, can readily do so using the open-source Python code contained in the Jupyter notebooks. For all users, we embed background material and references, prompts for further self-study, and problem set exercises to help them learn how to generate and adapt code. The exercises for each notebook focus on neuroscientific applications of the techniques being learned; thus, by working through the exercises, students learn how to use these techniques to answer meaningful neuroscientific questions (course instructors may contact us for more information).

To help users learn how to apply machine learning algorithms to cognitive neuroscience data, we build on several open-source machine learning tools in Python. For data loading we use Nibabel [28]; for data masking, normalization, dimensionality reduction, plotting, atlases, and functional connectivity we use Nilearn [19]; and for machine learning libraries we use Scikit-learn [29]. We include detailed instructions and exercises on how to avoid problems of circular inference and double-dipping. We also use tools native to BrainIAK for applying cutting-edge machine learning to fMRI data, including parallelized searchlight analysis [30]. An important consideration is how to prepare the data in a suitable format. Publicly available datasets are often in a raw state and need to be pre-processed (e.g., motion correction, registration, and masking) before they can be used for advanced analyses. The pre-processing can take a significant amount of time and add to the burden on the learner. To circumvent this bottleneck, we supply fully pre-processed data with the tutorials, making it significantly easier for a novice user to get started and quickly perform a successful analysis.

Having made it easy to access code and use machine learning algorithms, we embrace the third challenge: running the code efficiently using compute clusters. It can be difficult to take code that works on a laptop and modify it to efficiently leverage the resources of a cluster and scale performance to meet the demands of large datasets. This is a burden on the user and requires specialized expertise to write efficient, properly parallelized code. BrainIAK has built-in tools for making the most of clusters to scale analyses easily. In fact, the same code works seamlessly from a laptop (with a few cores) to clusters (with thousands of cores). For example, searchlight analysis (see [31]) involves running the same MVPA thousands of times at different points in the brain, which can be extremely slow on a laptop or desktop. BrainIAK includes a searchlight function that distributes these jobs on a cluster to run them in parallel. This function can be invoked using a few lines of code and runs seamlessly on any computing hardware. The tutorials give example code for cluster computing that can easily be extended to novel datasets.

In addition to parallelizing the code, cluster environments can present other complications for learners. In particular, the interactive nature of working on a laptop or desktop is absent

when working on a cluster, making troubleshooting difficult. Cluster environments also demand resource allocations up front (i.e., number of cores and amount of memory); increasing memory or extending time during program execution is not permitted. The tutorials use the SLURM scheduler [32] and provide instructions on how to determine the resources required to execute jobs and how to monitor running jobs.

In summary, we present a set of tutorials created to enable users of all skill levels to learn and deploy advanced multivariate fMRI analysis techniques. In addition to covering the latest incarnation of MVPA [1],[5], we provide recommendations on optimizing classifiers and strategies to avoid double-dipping. We also cover a range of cutting-edge techniques available in BrainIAK, including searchlight analysis, FCMA, ISC, ISFC, SRM, real-time fMRI, and event segmentation using hidden Markov models. We have released these tutorials publicly and freely. The users can also apply these methods to publicly available datasets from the existing literature, leading to independent validation of the published results. We are hopeful that this will help increase reproducibility of future results more broadly: when tutorial users analyze their own data, they will have already become familiar with the tools necessary to share their code and data, leading to a cycle of improved data sharing and code validation.

## Design and implementation

### Tutorials

Our learning materials are built and integrated using freely available tools and packages. The tutorials are written in the Python programming language. They are presented as Jupyter notebooks with background, documentation, and figures for each section of the code. For data loading, masking, and writing files in NIFTI format, we use Nibabel and Nilearn. A variety of functions useful for machine learning are called from Scikit-learn. Each notebook is paired with a publicly available dataset that is analyzed using the code (see Table 1). These datasets have already been pre-processed using standard steps and parameters, allowing the user to focus on learning the analyses. We have compiled a condensed version of these datasets, reducing the number of subjects to ensure that the tutorials run quickly and have reasonable memory requirements [33]. The results from the analyses are plotted using Matplotlib [34] and Seaborn [35]. For network connectivity diagrams, Networkx [36] and Nxviz were used. To load hdf5 files, the Deepdish package was used. The Watchdog package was used to indicate when new files were created.

### BrainIAK

BrainIAK is a software library for advanced fMRI analysis co-designed by cognitive neuroscientists and computer scientists. BrainIAK offers a Python interface and is mostly written in Python, but contains optimized code written in Cython and C++. Many of the methods in BrainIAK scale from a laptop to compute clusters using OpenMP [43] and MPI [44] parallel and distributed computing technologies. BrainIAK assumes that data have already been pre-processed with other pipelines and relies on other packages for plotting. The user is free to use any preprocessing pipeline (e.g., fmriprep, AFNI). Data are exchanged in standard NIFTI and NumPy formats with existing tools such as Nibabel or Nilearn and our tutorials show how to import data into Python structures and use BrainIAK. The functions in BrainIAK parse the data in a *time* x *voxels* format, with an exception being the searchlight function that takes in 4-D volumes. The BrainIAK package also serves as an ecosystem for users to contribute their own methods while avoiding duplication of methods found in other packages.

**Table 1. The datasets used in the tutorials.** We are releasing these datasets under the Creative Commons Attribution 4.0 International License. Although some of these datasets are publicly available, we provide condensed versions of these datasets, along with masks, that are easier to use and may be downloaded from Zenodo (https://doi.org/10.5281/zenodo.2598755). For quicker download speeds, the datasets may be downloaded from the Brainiak tutorials website (https:/brainiak.org/tutorials).

| | Datasets | Source | Used in tutorials | Online archive of the dataset |
|---|---|---|---|---|
| **1.** | Faces, places, and objects | [37] | 1–5, 7 | https://openneuro.org/datasets/ds001926/versions/1.0.1 <br> Condensed version available on the BrainIAK tutorials website. |
| **2.** | Ninety-six objects | [38] | 6 | Not on OpenNeuro. Condensed version available on the BrainIAK tutorials website. |
| **3.** | Faces and scenes | [39] | 7, 9 | Not on OpenNeuro. Condensed version available on the BrainIAK tutorials website. |
| **4.** | Lateralized attention | [40] | 8 | Not on OpenNeuro. Condensed version available on the BrainIAK tutorials website. |
| **5.** | Pieman story | [10] | 10, 11 | https://dataspace.princeton.edu/jspui/handle/88435/dsp015d86p269k <br> Condensed version available on the BrainIAK tutorials website. |
| **6.** | Raiders movie | [41] | 11 | https://github.com/HaxbyLab/raiders_data <br> Condensed version available on the BrainIAK tutorials website. |
| **7.** | Raiders images | [41] | 11 | https://github.com/HaxbyLab/raiders_data <br> Condensed version available on the BrainIAK tutorials website. |
| **8.** | Sherlock movie | [42] | 12 | https://openneuro.org/datasets/ds001132/versions/1.0.0 <br> Condensed version available on the BrainIAK tutorials website. |

## Hardware configurations

We have provided detailed instructions on how to configure the tutorials on different computing platforms here: https://brainiak.org/tutorials. Multiple installation or usage options are available using: Google Colaboratory for running through a web browser on the cloud, Docker for running on a Macintosh or Windows computer, and Conda for running on a Macintosh computer, Linux server, or high-performance compute cluster.

We tested the tutorials on clusters using the SLURM scheduler. We provide scripts to launch Jupyter notebooks on clusters and connect to the tutorials through a web browser via an SSH tunnel. We also provide bash scripts for running the tutorials on these remote servers. For long-running jobs that need large amounts of resources on the cluster, we use Python scripts that are submitted to the cluster as batch jobs instead of the more interactive Jupyter notebooks. These scripts are also provided along with the tutorials.

## Classroom deployment

These notebooks were initially developed for research methods courses taught at an advanced undergraduate/graduate level at Yale and Princeton. Each notebook was intentionally designed to be a suitable length for a weekly problem set that would take students between three and twelve hours, depending on the skill level of the student and complexity of the topic. To implement these tutorials in a classroom setting, we configured cluster resources for the class and distributed and collected assigned notebooks using GitHub Classroom. Another feature of GitHub Classroom is that it keeps student responses private from other students and yet gives the instructors easy access.

## Results

Our goal was to create user-friendly educational materials (https://brainiak.org/tutorials) that can be used by novice or expert practitioners to learn how to deploy advanced fMRI analyses in their research. The execution of the notebooks on a cluster is also made simple. If the requisite software and data are installed on the cluster, a user simply needs to connect to the cluster from their laptop/desktop computer, open a web browser, and access the Jupyter notebooks. The tutorials can also be run on the cloud for free via Google Colaboratory. Each tutorial notebook has an overarching theme of a scientific question relevant to cognitive neuroscience. The

accompanying notebook exercises help the user understand the method and its applicability to the scientific question by requiring that they generate answers or code. The questions and exercises can be used to formally evaluate students enrolled in a for-credit course (course instructors may contact us for more information). These questions are posed in the context of a publicly available fMRI dataset. These datasets are distributed with the tutorials in a ready-to-use (pre-processed) state. The user is also encouraged to make novel contributions using the method that they learned in the tutorial, either by enhancing the method, creating a new visualization of the data, or even using the method on another dataset, e.g., from OpenNeuro (http://openneuro.org) [45]. Once the user has acquired proficiency in executing the notebooks from a browser, we introduce running programs on clusters by submitting scripts as batch jobs.

Each of the notebooks can be run independently. For the beginning and intermediate user, we recommend starting with the first notebook and working through 1–7. After this, the user can choose to focus on a particular method among notebooks 8–13. An advanced user already familiar with Python and machine learning can start with any notebook in the sequence. For those who are new to clusters but otherwise proficient at fMRI analysis, the searchlight notebook is a useful starting point. We describe the contents of each notebook (https://brainiak.org/tutorials) in more detail below:

## Tutorial notebooks

1. **Setup:** An introductory notebook to help users learn how to work with Jupyter.

2. **Data handling and normalization:** Load fMRI datasets into a Python environment using Nilearn and Nibabel packages. The importance of normalizing the data is shown via an exercise using a simulated dataset.

3. **Classification:** Once the data have been loaded and normalized, the BOLD signal is extracted with a shift to account for hemodynamic lag and classification is performed using a linear classifier. The importance of separating training and test data is emphasized and cross-validation is introduced. The pitfalls of double-dipping are highlighted and the leave-one-run-out approach is covered. A category localizer dataset is used to examine modular vs. distributed processing in the visual system.

4. **Dimensionality reduction:** Introduce principal component analysis (PCA), explore how to select the number of dimensions, and highlight the importance of using cross-validation to perform feature selection. Determine the smallest number of components yielding the "best" decoding accuracy. Show how other dimensionality reduction techniques can be substituted into this pipeline.

5. **Classifier optimization:** Use grid search and pipelines from Scikit-learn to tune hyperparameters and perform nested cross-validation. How to handle mild forms of double-dipping (e.g., "peeking" at unlabeled test data by including it in z-scoring) that are often unavoidable, by performing permutation tests with randomized labels.

6. **Representational similarity analysis (RSA):** Using pattern similarity and representational dissimilarity matrices to explore the neural representation of different categories of objects in a way that can be compared to behavioral judgments and computational models, and solve the identity of unlabeled "mystery" objects.

7. **Searchlights:** Explore where in the brain local areas contain multivariate information that discriminates between faces and scenes. Begins with a small mask to build proficiency and

ends by running a whole-brain searchlight analysis. Demonstrates how to execute this computationally intensive analysis rapidly on a cluster using batch scripts and covers resource planning and monitoring of large batch jobs.

8. **Seed-based functional connectivity:** To explore how large-scale brain networks, not just individual regions, contribute to cognitive processing, examine the temporal correlation (functional connectivity) between regions. Shows how connectivity changes during an attention task and how to remove stimulus-evoked responses to isolate background connectivity.

9. **Full correlation matrix analysis (FCMA):** Rather than focus on connectivity with one or more seed regions of interest, calculate and analyze an unbiased measure of connectivity—the correlation of every voxel in the brain with every other voxel. Highlights differences between FCMA (which classifies based on connectivity) and MVPA (which classifies based on activity), including brain regions that are equally active for faces and scenes but are differentially connected.

10. **Inter-subject connectivity (ISC):** Examine what is common across people by measuring correlations over time in the activity of matching voxels in their brains in response to a common stimulus (e.g., story or movie). Measure functional connectivity across people by correlating non-matching voxels (e.g., between angular gyrus in one subject and hippocampus in another). Shows how these techniques can reveal stimulus-driven variance in the brain by comparing listening to intact vs. scrambled stories.

11. **Shared response model (SRM):** A common stimulus across subjects can be used to align subject brains functionally, rather than typical anatomical registration. SRM seeks to find shared variance in the fMRI data across subjects, in a reduced dimension feature space. This results in weights that map between voxels and features, allowing other data to be projected into the aligned space. SRM can also be viewed as a technique for isolating reliable stimulus-related responses by removing responses that are either noise or idiosyncratic subject responses. Shows the utility of this approach by improving time-segment matching in movie data and image classification with MVPA.

12. **Event segmentation:** Use hidden Markov models (HMMs) to identify a sequence of transitions between stable brain patterns in fMRI data. Illustrates how fitting HMMs to data from high-level brain regions (obtained during movie-watching) subdivides the time series into chunks that track events in the movie. Explores whether retrieving events from memory leads to similar neural transitions.

13. **Real-time fMRI:** Most fMRI studies involve collecting data and analyzing them days or weeks later. By analyzing data on the fly, real-time fMRI makes new kinds of experiments possible, such as neurofeedback training and adaptive designs. Demonstrates the use of an fMRI data simulator, which generates brain images at the rate of an fMRI study (every 1–2 s), and then address how to pre-process data online and how to complete MVPA or other advanced analyses incrementally, before the next brain image.

## Cluster computing

Analyses that require either a long run-time or large memory are best run in batch mode. The Jupyter notebooks for these jobs serve as a template and may be used as the starting point for a batch script. Once the contents of the notebook have been learned, the user is directed to execute batch scripts associated with the notebook on the cluster.

Executing batch jobs on clusters is non-trivial as it involves allocating the correct memory utilization, number of tasks, and the time required. Given the non-interactive nature of most clusters, debugging performance issues can be challenging. In the Searchlight notebook we have provided step-by-step instructions for cluster execution. To make the transition to running on clusters easier, we provide recommendations such as running small samples of the analyses and extrapolating to make memory and time estimates for the analysis of the entire dataset. We also provide batch scripts with parameters that can be changed to fit the needs of the user. Finally, we provide some basic tips on how to monitor the status of batch jobs on the clusters.

## Other resources

To use the tutorials, a user will need to interact with multiple software tools. To make it easier for a new user to navigate these tools, we have created a website https://github.com/brainiak/brainiak-tutorials/wiki/Resources, where a new user can access tutorials and become familiar with Python, GitHub, and Unix. Furthermore, our goal for these tutorials was to cover advanced fMRI analysis and hence our tutorials do not cover pre-processing methods, General Linear Model analysis, or software deployment options (e.g., containers) in great detail. An exhaustive list covering multiple helpful tools and tutorials is available here: https://github.com/ohbm/hackathon2019/blob/master/Tutorial_Resources.md.

## Availability and future directions

These tutorials and their associated datasets can be accessed here: https://brainiak.org/tutorials. At the time of writing there are 13 notebooks available. As time permits, we intend to produce more tutorials as needs or new methods demand. The methods/tools that are available in BrainIAK but not covered in the tutorials are: Bayesian derived methods for RSA; Topographic Factor Analysis; and an fMRI Simulator. We welcome contributions to BrainIAK from the community, in the form of code and tutorials added via GitHub.

## Acknowledgments

We would like to thank the following people for help with many aspects of the tutorials: David Turner for help with high-performance computing; Benjamin Singer for help with software installations and data management; Grant Wallace for cloud configurations and testing; and Daniel Suo for website management. Several individuals contributed to specific tutorials, as listed in the contributions section for each tutorial; we especially thank Chris Baldassano for creating the initial HMM notebook and writing an example script to compute ISC, and Po-Hsuan (Cameron) Chen for providing initial code for the SRM notebook. We would also like to thank the students (at Yale and Princeton) and workshop and hackathon participants (at Yale, Princeton, and Virginia Tech) for their participation and feedback on these materials, Ed Clayton for organizing logistics for the workshops and hackathons, and Jonathan D. Cohen for overall project oversight. The author order for N.B.T.-B and K.A.N was determined by a coin flip.

## Author Contributions

**Conceptualization:** Manoj Kumar, Cameron T. Ellis, Peter J. Ramadge, Nicholas B. Turk-Browne, Kenneth A. Norman.

**Software:** Manoj Kumar, Cameron T. Ellis, Qihong Lu, Hejia Zhang, Mihai Capotă, Theodore L. Willke.

**Supervision:** Theodore L. Willke, Peter J. Ramadge, Nicholas B. Turk-Browne, Kenneth A. Norman.

**Writing – original draft:** Manoj Kumar.

**Writing – review & editing:** Manoj Kumar, Cameron T. Ellis, Qihong Lu, Hejia Zhang, Mihai Capotă, Peter J. Ramadge, Nicholas B. Turk-Browne, Kenneth A. Norman.

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
