## [Decision Letter · Decision Letter 0]

26 Aug 2019

Dear Dr Kumar,

Thank you very much for submitting your manuscript 'BrainIAK tutorials: user-friendly learning materials for advanced fMRI analysis' for review by PLOS Computational Biology. Your manuscript has been fully evaluated by the PLOS Computational Biology editorial team and in this case also by independent peer reviewers. 

Your paper describes a valuable contribution, and the examples are comprehensive. 

The reviewers have raised some important points concerning the implementation of the tutorials (in addtition to the feedback you already received by the community), and the inclusion of this set of tools in the panorama of a wider communitary effort.

While your manuscript cannot be accepted in its present form, we are willing to consider a revised version in which the issues raised by the reviewers have been adequately addressed. We cannot, of course, promise publication at that time.

Sincerely,

Daniele Marinazzo

Deputy Editor

PLOS Computational Biology

Daniele Marinazzo

Deputy Editor

PLOS Computational Biology

[LINK]

Reviewer's Responses to Questions

**Comments to the Authors:**

Reviewer #1: Review uploaded as attachment

Reviewer #2: By drafting the described tutorials, the authors have provided a substantial, valuable contribution to the field. I do, however, have significant concerns about their framing of this contribution in the present manuscript. In particular, the authors fail to acknowledge related efforts across open neuroimaging. I hope that this review provides some constructive feedback as to where they could better link their efforts with those of other community members so that a reader might better understand the impact of the described tutorials.

The full review is uploaded as an attachment.

Reviewer #3: # Summary and general comments

In this submission, Kumar and colleagues present a library called BrainIAK for machine learning in functional neuroimaging, and an accompanying set of tutorials.

The tutorials are presented in the form of jupyter notebooks, and are accessible either locally through containers or online on the google collab platform.

They also include instructions for deployment on high-performance infrastructure.

The data used in the tutorial are freely available and specially prepared to be used as part of a training activity.

As a strength, some of the material covered in the tutorials include inter-subject correlations and representational similarity analysis, two applications which are not well covered by currently available tutorials, to my knowledge. Overall, this new library and tutorials are remarkably comprehensive, and I believe will represent a very valuable resource for the community. My only major concern is that the authors did not properly position their work compared to other efforts.

# Minor comments

* abstract: citing a specific list training and hackathons will become obsolete in a few months only. Maybe stay vague there.

* intro claims several times the lack of existing education material. There is a huge amount of general-purpose tutorials for machine learning, most notably featuring the sklearn documentation. There are at least three extensive packages with many tutorials: nilearn, pyMVPA and https://www.ncbi.nlm.nih.gov/pmc/articles/PMC4956688/. The authors should briefly review these other resources and explain how BrainIAK adds to them (paragraph line 120). Which techniques are not currently covered by tutorials?

* l. 211: would you have recommendations for resources to preprocess the data that would integrate well with BrainIAK? In particular, you may want to discuss if detailed instructions are available for importing minimally preprocessed data, such as the ones generated by fMRIprep.

* no material is presented to demonstrate that the proposed material achieves the stated goals. Survey results from a workshop, for example, would add some support to the usefulness of the resources.

# Optional suggestions

Below are two suggestions. I (as a reviewer) do not think it is necessary to implement these suggestions prior to publication. I am providing these suggestions in the hope the authors may find them useful and may choose to follow up on some of them.

* BrainIAK should go through a proper code review, as a library. Consider a submission to the journal of open source software (JOSS) for the library component of BrainIAK.

* I have not reviewed the tutorials themselves, but tried to evaluate if BrainIAK adds conceptually to existing software resources. As part of the NeuroLibre platform, a detailed technical review of the notebooks has been performed by two reviewers. I would encourage the authors to address these technical issues.

**Have all data underlying the figures and results presented in the manuscript been provided?**

Reviewer #1: Yes

Reviewer #2: Yes

Reviewer #3: Yes

PLOS authors have the option to publish the peer review history of their article (what does this mean?). If published, this will include your full peer review and any attached files.

Reviewer #1: Yes: Oscar Esteban

Reviewer #2: No

Reviewer #3: Yes: Pierre Bellec

---

## [Decision Letter · Decision Letter 1]

17 Nov 2019

Dear Dr Kumar,

We are pleased to inform you that your manuscript 'BrainIAK tutorials: user-friendly learning materials for advanced fMRI analysis' has been provisionally accepted for publication in PLOS Computational Biology. Please make sure to update the table and references as requested by reviewer 2.

In the meantime, please log into Editorial Manager at https://www.editorialmanager.com/pcompbiol/, click the "Update My Information" link at the top of the page, and update your user information to ensure an efficient production and billing process.

One of the goals of PLOS is to make science accessible to educators and the public. PLOS staff issue occasional press releases and make early versions of PLOS Computational Biology articles available to science writers and journalists. PLOS staff also collaborate with Communication and Public Information Offices and would be happy to work with the relevant people at your institution or funding agency. If your institution or funding agency is interested in promoting your findings, please ask them to coordinate their releases with PLOS (contact ploscompbiol@plos.org).

Thank you again for supporting Open Access publishing. We look forward to publishing your paper in PLOS Computational Biology.

Sincerely,

Daniele Marinazzo

Deputy Editor

PLOS Computational Biology

Daniele Marinazzo

Deputy Editor

PLOS Computational Biology

<br \\>

Reviewer's Responses to Questions

**Comments to the Authors: **

Reviewer #1: Please find my comments attached.

Oscar Esteban.

Reviewer #2: The authors have addressed my major concerns, and the revised manuscript significantly better situates these contributions in the context of the broader field. Several minor notes and clarifications:

1. I am delighted that the authors are careful to cite supporting software, but I noticed that two technologies are missing from the references list: Jupyter Notebooks and OpenNeuro (formerly OpenfMRI). These citations are, respectively: 

Kluyver, T., Ragan-Kelley, B., Pérez, F., Granger, B.E., Bussonnier, M., Frederic, J., Kelley, K., Hamrick, J.B., Grout, J., Corlay, S., Ivanov, P., Avila, D., Abdalla, S., Willing, C., & Jupyter development team (2016). Jupyter Notebooks - a publishing format for reproducible computational workflows. In Positioning and Power in Academic Publishing: Players, Agents and Agendas. doi: 10.3233/978-1-61499-649-1-87.

and

Poldrack, R.A., Barch, D.M., Mitchell, J.P., Wager, T.D., Wagner, A.D., Devlin, J.T., Cumba, C., Koyejo, O., and Milham, M.P. (2013). Toward open sharing of task-based fMRI data: the OpenfMRI project. Frontiers in Neuroinformatics, 7, 1–12.

Could the authors please update the text to include these references?

2. I appreciate the authors' clarification as to why Google drive links were included for the datasets. I was also pleased to see that the data used in the tutorials are now directly available in Zenodo, as this provides better long-term archiving. 

Would the authors be willing to update their caption for Table 1 to directly link to the Zenodo archive? This would ensure better long-term access to the exact data source used in the tutorials, as the authors point out that the versions available from e.g. OpenNeuro do not match those used in the lessons.

Reviewer #3: Thanks for appropriately addressing all of my comments, and congratulations on a very valuable contribution.

**Have all data underlying the figures and results presented in the manuscript been provided?**

Reviewer #1: Yes

Reviewer #2: Yes

Reviewer #3: Yes

PLOS authors have the option to publish the peer review history of their article (what does this mean?). If published, this will include your full peer review and any attached files.

<br \\>

<br \\>

Reviewer #1: Yes: Oscar Esteban

Reviewer #2: No

Reviewer #3: Yes: Pierre Bellec

---

## [Editor Report · Acceptance letter]

9 Dec 2019

PCOMPBIOL-D-19-01130R1 

BrainIAK tutorials: User-friendly learning materials for advanced fMRI analysis

Dear Dr Kumar,

I am pleased to inform you that your manuscript has been formally accepted for publication in PLOS Computational Biology. Your manuscript is now with our production department and you will be notified of the publication date in due course.

With kind regards,

Laura Mallard
